# Diurnal Physical Activity Patterns across Ages in a Large UK Based Cohort: The UK Biobank Study

**DOI:** 10.3390/s21041545

**Published:** 2021-02-23

**Authors:** Julia Wrobel, John Muschelli, Andrew Leroux

**Affiliations:** 1Department of Biostatistics and Informatics, Colorado School of Public Health, University of Colorado Anschutz Medical Campus, Aurora, CO 80045, USA; ANDREW.LEROUX@CUANSCHUTZ.EDU; 2Department of Biostatistics, Bloomberg School of Public Health, Johns Hopkins University, Baltimore, MD 21218, USA; jmusche1@jhu.edu

**Keywords:** accelerometers, aging, UK Biobank, functional regression, curve registration

## Abstract

The ability of individuals to engage in physical activity is a critical component of overall health and quality of life. However, there is a natural decline in physical activity associated with the aging process. Establishing normative trends of physical activity in aging populations is essential to developing public health guidelines and informing clinical perspectives regarding individuals’ levels of physical activity. Beyond overall quantity of physical activity, patterns regarding the timing of activity provide additional insights into latent health status. Wearable accelerometers, paired with statistical methods from functional data analysis, provide the means to estimate diurnal patterns in physical activity. To date, these methods have been only applied to study aging trends in populations based in the United States. Here, we apply curve registration and functional regression to 24 h activity profiles for 88,793 men (N = 39,255) and women (N = 49,538) ages 42–78 from the UK Biobank accelerometer study to understand how physical activity patterns vary across ages and by gender. Our analysis finds that daily patterns in both the volume of physical activity and probability of being active change with age, and that there are marked gender differences in these trends. This work represents the largest-ever population analyzed using tools of this kind, and suggest that aging trends in physical activity are reproducible in different populations across countries.

## 1. Introduction

Physically active individuals have a longer life expectancy and increased health span [1,2,3]. Because physical activity (PA) levels are modifiable for most adults, PA is an attractive target for interventions aimed at improving the quality of life in older adults. In addition, features of PA are known to be highly correlated with prevalence of various health conditions [4] and risk of mortality [5,6,7,8,9]. Taken together, these observations suggest the power of monitoring PA in a free-living environment to both inform the epidemiology of healthy aging and facilitate safe, independent, home living for aging individuals if incorporated into, for example, a clinical monitoring program through individuals’ primary care provider. Historically, PA has been most often measured using self-report questionnaires, which are prone to substantial biases [10]. Wearable accelerometers provide a convenient, non-invasive, objective alternative for measuring PA, and have become widely adopted in health studies such as the Baltimore Longitudinal Study on Aging (BLSA) [11], the National Health and Nutrition Survey (NHANES) 2003–2006 and 2011–2014 [12], and the UK Biobank [13]. Moreover, in the context of aging, the ability to collect objective measures of physical activity are crucial due to the increasing prevalence of cognitive deficiencies which can further bias self-reported levels of PA.

The increased use of accelerometers in observational studies, clinical trials, large biobanks, and for recreational purposes has provided a wealth of data that can be used to obtain objective measurements of physical activity in free-living environments. Accelerometers typically measure acceleration in three orthogonal axes at the sub-second level, capturing high resolution information on horizontal, lateral, and vertical movement. Typically, these sub-second level data are aggregated at lower resolution time intervals called epochs, most commonly 1 min epochs; this produces multiple days of 24 h minute-by-minute activity trajectories for each subject. When combined across days to create a single 24 h trajectory, these are referred to as daily acceleration or activity profiles. Most often, analysis of data generated by accelerometers focuses on creating one number summaries of the daily profiles for each subject, typically measuring volume of PA (e.g., step count, sedentary time, active time, etc.), circadian rhythm (e.g., relative amplitude), or features of sleep (e.g., sleep efficiency, sleep duration, number of wakes). Summarizing multiple days of data using this approach ignores the highly correlated nature of these features and fails to efficiently exploit information contained in the timing of PA. To overcome this limitation, recent work by Di et al. [14] used the joint and individual variation explained approach [15] to characterize patterns of PA, circadian rhythms, and sleep using a set of scalar features from each domain. An alternative approach involves analyzing the entire 24 h acceleration profiles in conjunction with health outcomes. Despite evidence that specific patterns of timing and magnitudes of physical activity over the 24 h day are associated with aging [11] and mortality [7,16], analytic approaches that use the full acceleration profiles have been underutilized in the literature, perhaps because of the computational and methodological challenges of working with high dimensional, correlated, structured data.

Fortunately, statistical methods developed in the field of functional data analysis [17] provide a natural framework for analyzing acceleration profiles. From the functional data perspective, each 24 h acceleration profile is a statistical unit of observation that can be analyzed using methods specifically developed to extract patterns of variation and perform inference on noisy, dense, correlated data [18,19]. Here we focus on analyzing diurnal patterns of physical activity using two methods from functional data analysis. Specifically, we use function-on-scalar regression (FoSR) [20], a method which treats an entire “function” (acceleration profile) as an outcome and associates the entire function with scalar features (e.g., age), and curve registration [21] separates 24 h activity profiles into components of “horizontal” variability and “vertical” variability which correspond to timing and magnitude of PA, respectively. Function-on-scalar regression has been used to analyze age-associated trends in diurnal patterns of physical activity in the BLSA [22], though that analysis may not scale up to massively large datasets such as the UK Biobank. In addition, the study using BLSA data employed a single axis accelerometer, worn at the chest, and the unit of measurement analyzed was created using a proprietary algorithm which is not transferable across devices. Our analysis is based on open-source and published algorithms using a tri-axial wrist worn accelerometer, which should allow for more general use in other studies. As a result, it is unclear whether and how those analyses would be replicated using a tri-axial accelerometer placed on a different location on the body, and using a different unit of measurement. Though registration of acceleration profiles was recently used to uncover sub-types of circadian rhythms [23], to our knowledge registration has never been used to study how trends in circadian rhythms change with age, nor has it been paired with functional regression methods. We focus here on functional regression for continuous and binary data, and combine FoSR with curve registration for binary data to simultaneously analyze different aspects of sex-specific age trends in diurnal patterns of PA, leveraging the information contained in the complex high dimensional acceleration profiles to draw novel insights about diurnal patterns in PA across ages.

Studying PA across a wide span of age ranges, especially at older ages, typically requires a large number of participants. Opportunely, the UK Biobank is a large prospective cohort study that enrolled more than 500,000 adults. In the UK Biobank accelerometer sub-study, over 100,000 adults wore a wrist-worn accelerometer (Axtivity AX3, Newcastle upon Tyne, UK) for 7 days [24] between the years of 2013 and 2015. Approximately 88% of those enrolled report British ancestry. The UK Biobank collected vast amounts of information regarding participants’ socio-demographic, lifestyle, environment, accelerometry, imaging and genetics [25,26]. In addition, participants’ data can be linked to incident morbidity and mortality through hospital and death records. With data of this size, we can potentially uncover previously undetectable patterns in PA across ages and assess whether diurnal aging trends observed in US based cohorts are replicated in a large UK cohort study.

This work aims to extend the existing literature on objectively measured physical activity using wearable devices in older adults in several ways. First, we use an open source, reproducible summary of raw sub-second level acceleration data aggregated at the minute level to assess the replicability of the general sex-specific trends and differences in the timing and volume of PA observed in US based populations in a large UK based prospective cohort study. Specifically, we wish to validate sex-specific differences in the time-of-day trends these trends using milli-gravity units values as opposed to previously-published proprietary activity counts. Second, we illustrate different patterns of activity of ages if using the values of activity versus an indicator of active vs. inactive using thresholded data. Third, we introduce the concept of curve registration to the intersecting field of physical activity and aging as a tool for analyzing epidemiological trends. Fourth, we show that complex functional regression methods are computationally feasible on very large physical activity datasets using high quality open source software. Finally, to further the goal of the dissemination of our functional data methods applied to physical activity data we provide an accompanying vignette that fully reproduces our analysis in the NHANES 2003–2006 accelerometry data. To avoid the potential complications associated with differential weekend versus weekday patterns of activity, we focus here only on weekdays (Monday–Friday).

## 2. Materials and Methods

### 2.1. Data and Preprocessing

#### UK Biobank: Accelerometry Sub-Study

The UK Biobank accelerometry data were collected between June 2013 and December 2015, and participants wore the devices an average of 5.7 years (range 2.8 to 9.7 years) after the date of their initial assessment. Doherty et al. [24] describes the inclusion criteria, sampling design, and other aspects of the UK Biobank accelerometry sub-study. In summary, individuals wore a wrist-worn accelerometer for up to 7 days, providing acceleration values in milli-gravity units (g = 9.81$\frac{m}{s^{2}}$). The data are made available at varying resolutions: (1) raw sub-second level tri-axial data; (2) 5-second level aggregated acceleration; and (3) subject-level summaries averaged across days. The 5-second level aggregated data is created by taking the Euclidean norm of the time-series data from the 3 axes, subtracting 1 gravitational unit for gravity, then averaging within 5-second intervals (called ENMO for Euclidean Norm minus one). Doherty et al. [24] describes the exact procedure for deriving the 5-second level data from the raw data and estimating periods of non-wear. We use the 5-second level epoch data because the subject-level summaries do not provide day-level information required for our analysis, and reprocessing the raw accelerometry data is a vast undertaking which is beyond our scope.

We follow a recent study from Leroux et al. [9] to aggregate the 5-second level data into 1-minute epochs. This aggregation involves averaging the 5-second level data combined with a step to impute the small proportion of missing data in the sample. A detailed description of this procedure is provided in the supplemental material of Leroux et al. [9]. We apply the same exclusion criteria as [9] to determine what constitutes a “good” day of accelerometry data. Specifically, we determine a good day of acceleration data to be one with sufficient estimated wear time (≥95% of the day, or 1368 min), as well meeting the three data quality criteria provided by the UK Biobank, which relate to data calibration and sufficient device wear time. The exact criteria can be found at the UK Biobank data showcase website (https://biobank.ndph.ox.ac.uk/showcase/ accessed on 1 January 2021) by searching the field IDs 90015, 90016, and 90017. In addition, we require individuals to have at least 3 good weekdays (Monday–Friday) of accelerometry data. We included all individuals who met these criteria.

After applying these inclusion criteria, our analytic sample contained 88,793 participants who were an average age 61.9 (range 42.8–78.7) at the time of accelerometer wear and 49,538 (55.8%) of whom were female. In addition, 16.1%, 66.0%, and 17.9% of participants had 3, 4, and 5 good days of weekday accelerometry data, respectively. There is evidence [9] that individuals who participated in the UK Biobank accelerometry study were younger and healthier than the UK Biobank study as a whole, which has been reported to be healthier than the general UK population [27]. As a result, our analytic sample is likely subject to multiple sources of healthy participant bias.

### 2.2. Methodology and Analysis

#### Physical Activity Acceleration Profiles

Each individual has 24 h trajectories of minute-level acceleration data for multiple days that are condensed into physical activity (acceleration) profiles that represent one day of activity for each subject. We consider two types of activity profiles that confer complementary benefits and interpretations. First, let $Z_{ij}\left(t\right)$, represent the observed ENMO for participant *i* on day *j* at every minute t=1,…,1440. We collapse these $Z_{ij}\left(t\right)$ across days where each individual has $J_{i}$ days of good data with $3\leq J_{i}\leq 5$ based on adherence to wear-time protocols during the weekdays as
$$Z_{i}\left(t\right)=J_{i}^{-1}\sum_{j=1}^{J_{i}}Z_{ij}\left(t\right)\;,$$
and call these $Z_{i}\left(t\right)$ continuous activity profiles. Our second type of activity profile is denoted $Y_{i}\left(t\right)$ and represents subjects’ median tendency to be have observed ENMO above or below a threshold of 30 milli-gravity units. Here we use 30 milli-gravity units as this is a candidate threshold of active versus sedentary behaviors. We refer to this threshold as a “candidate threshold” for active versus sedentary behaviors as it has been used in practice [9], but has not yet been widely validated, though hereafter we will refer to this as simply active versus inactive behaviors for simplicity. Specifically, let
$$Y_{i}\left(t\right)=\lceil \mathrm{med}(\left\{1\left(Z_{ij}\left(t\right)>=30\right):j=1,\ldots ,J_{i}\}\right\rceil \;,$$
where the ceiling function ⌈⌉ rounds up the median value across days in the event that $J_{i}$ is odd and the median is 0.5. These $Y_{i}\left(t\right)$, which we denote binary activity profiles, are binary trajectories representing active ($Y_{i}\left(t\right)=1$) and inactive minutes ($Y_{i}\left(t\right)=0$) over the 24 h period. In short, $Z_{i}\left(t\right)$ is the average daily profile and $Y_{i}\left(t\right)$ is the indicator if the majority of days have activity in this minute (above 30 milli-gravity units).

There are two main motivations for constructing these two different profiles. First, although registration is possible on the acceleration data itself, $Z_{i}\left(t\right)$, short bursts of high intensity exercise may skew activity counts and exert undue influence on registration results and this effect is attenuated by using binary profiles $Y_{i}\left(t\right)$ [23]. Second, it is possible that these two profiles contain different pieces of information about how activity patterns change with age. For example, it may be that the likelihood of being active does not change much with age, but that the volume of activity is much less in older adults. Thus, we choose to analyze the changes in both of these activity profiles as a function of age.

#### Curve Registration

Our registration procedure is adapted from Wrobel et al. [21] for the UK Biobank accelerometer data and aligns binary activity profiles by common features such as wake time, sleep time, and peaks in probability of being active. An example of registration is shown in Figure 1: binary activity profiles (Figure 1A) are used to estimate $\mu_{i}\left(t\right)$, the probability of subject *i* being active at time *t* (Figure 1B), then warping functions (Figure 1C) stretch and compress the time domain to produce aligned probability curves and activity profiles (Figure 1D).

After registration, horizontal variability contained in the warping functions provide information about timing of PA, and vertical variability contained within the aligned activity profiles provide information about magnitude of PA; both the warping functions and binary activity profiles will be analyzed in Section 3. Notation and model details for this process are given below, and for a full technical explanation we refer the reader to Wrobel et al. [21].

Let *t* represent observed chronological time. Then $Y_{i}\left(t\right)$ and $\mu_{i}\left(t\right)$ represent the binary activity profile and activity probability curves, respectively over chronological time for the *i*th subject. The warping functions, denoted $h_{i}^{-1}$ for subject *i*, nonlinearly stretch and compress chronological time to define a new time domain $t^{\prime }=h_{i}^{-1}\left(t\right)$ on which activity profiles are registered. This registration occurs via a two-step iterative algorithm where in step (1) the probability curves $\mu_{i}\left(t\right)$ are estimated using binary functional principal component analysis (FPCA) and in step (2) the warping functions $h_{i}^{-1}\left(t\right)$ are estimated via constrained likelihood maximization. The model for this joint FPCA-registration algorithm is
$$\begin{array}{cc} E\left[Y_{i}\left(h_{i}^{-1}\left(t\right)\right)|c_{i},h_{i}^{-1}\right] & =\mu_{i}\left(t^{\prime }\right)\\ logit\left[\mu_{i}\left(t^{\prime }\right)\right] & =\alpha \left(t^{\prime }\right)+\sum_{k=1}^{K}c_{ik}\psi_{k}\left(t^{\prime }\right), \end{array}$$
where $Y_{i}\left(h_{i}^{-1}\left(t\right)\right)=Y_{i}\left(t^{\prime }\right)$ represent the registered activity profiles, $\alpha (t^{\prime })$ is the population mean function, and $c_{ik}$ and $\psi_{k}\left(t^{\prime }\right)$ are the subject-specific scores and population-level eigenfunctions, respectively, for the *k*th principal component. The algorithm iterates between step 1 and step 2 until activity profiles are aligned, resulting in estimated warping function $h_{i}^{-1}\left(t\right)$ for each participant.

#### Functional Regression

We fit a series of functional regression models that address the following question: what is the epidemiology of activity patterns with age in the UK Biobank and how does disentangling horizontal and vertical variability via registration amplify or modify these trends? We choose to model these associations flexibly using techniques from generalized function-on-scalar regression (FoSR) [28,29], where we allow for the association between the outcome to vary smoothly in time of day, *t*, and age on the linear predictor scale. We fit separate models for: $Y_{i}\left(t\right)$, $Z_{i}\left(t\right)$, and $h_{i}^{-1}\left(t\right)$ stratified by sex where $h_{i}^{-1}\left(t\right)$ are the participant-specific estimated warping functions obtained from the registration algorithm described in Section 2.2. Specifically, we fit models of the form
$$g\left(E\left[Y_{i}\left(t\right)|{\mathrm{Age}}_{i}\right]\right)=f_{0}\left(t\right)+f_{1}\left(t,{\mathrm{Age}}_{i}\right)$$
where g(·) is a link function and $f_{0}\left(t\right),f_{1}\left(t,{\mathrm{Age}}_{i}\right)$ are unspecified smooth functions. When modelling $Z_{i}\left(t\right)$ and $h_{i}^{-1}\left(t\right)$ we specify *g* to be the identity function and for $Y_{i}\left(t\right)$ we specify *g* be the logit function. For identifiability of $f_{1}$ we impose the constraint $\sum_{i=1}^{N}f_{1}\left(t,{\mathrm{Age}}_{i}\right)=0$ for all *t*. Note that the method we use for estimating an age effect can be easily extended to account for other covariates such as body mass index (BMI), comorbidities, and lifestyle factors. In addition, the specification of the model can be simplified to allow terms which vary linearly in the covariate (i.e., $f_{1}\left(t,{\mathrm{Age}}_{i}\right)=f_{1}\left(t\right)\times {\mathrm{Age}}_{i}$), also referred to as a varying coefficient model [30], or to vary smoothly in the covariate but linearly in time (i.e., $f_{1}\left(t,{\mathrm{Age}}_{i}\right)=f_{1}\left({\mathrm{Age}}_{i}\right)\times t$), or to be linear in the level of the covariate and fixed over time (i.e., $f_{1}\left(t,{\mathrm{Age}}_{i}\right)=\beta_{1}{\mathrm{Age}}_{i}$). Further note that we fit these models stratified by sex, though a unified model could be fit by specifying $g\left(E\left[Y_{i}\left(t\right)|{\mathrm{Age}}_{i}\right]\right)=f_{0}\left(t\right)+f_{1}\left(t,{\mathrm{Age}}_{i}\right)+f_{2}\left(t,{\mathrm{Age}}_{i}\right){\mathrm{Sex}}_{i}$ where ${\mathrm{Sex}}_{i}$ is an indicator for whether participant *i* is female.

The two primary methodologic challenges in fitting FoSR models are estimation of smooth fixed effects and accounting for within-subject correlation. As our interest here is in the estimation of population-level marginal models, we take a a bootstrap procedure for both estimation and inference on fixed effects [31]. We use cyclic cubic regression splines for estimating $f_{0}$, and a tensor product smooth of marginal cyclic cubic splines and cubic splines for estimating $f_{1}$. Our use of cyclic splines in the time direction is motivated by the fact that time of day is cyclical, so the estimated coefficient should meet at 12 a.m. We penalize the curvature of the coefficients using a second derivative penalty to avoid over-fitting of the model to the data, the degree of the curvature is controlled by a tuning parameter referred to as a “smoothing parameter”. Methods for automatic smoothing parameter selection are a key challenge in functional regression methodology based on penalized splines. Here, smoothing parameter selection is done using the fast REML procedure described in [32].

#### Computation, Software, and Reproducibility

We have 88,793 participants in our analytic sample with 1440 observations per subject, resulting in a massive 127,861,920 total observations. We were able to fit our functional regression models in a reasonable computation time of 11.1 min (0.19 h) and 72.2 min (1.20 h) for continuous and binary outcomes, respectively, using smoothing software optimized for very high dimensional data[32]. For registration we parallelized the estimation of subject-specific warping functions using a node on a high performance computing cluster with 50 cores. In total, the registration step took 24.0 h to complete. All methods are implemented in the statistical software *R* [33]. Functional regression models are implemented using the mgcv::bam() function in the mgcv [34] package, and registration is performed using the registr package [35].

The UK Biobank data are publicly available, but access requires approval on a project with specific aims. Moreover, individual identifiers are randomized for each project such that individuals cannot be directly linked between projects. As a result, reproducibility of our results in the UK Biobank is limited. In the interest of disseminating our work, we provide code for performing all of our analyses using the publicly accessible accelerometry data from the National Health and Nutrition Survey 2003–2006 [36] which has been organized in an analytic ready format through the R package rnhanesdata [7,37]. Our analysis is reproduced using the NHANES data via a supplementary markdown file to be uploaded to Github upon publication.

### 2.3. Populations of Comparison

We compare the results of our analysis on the UK Biobank 2013–2015 accelerometry study to similar analyses performed in the NHANES 2003–2006 data [38] and the BLSA [22]. The accelerometry data used in the BLSA study was collected between August 2007 and January 2011 [11]. The NHANES is a nationally representative sample of the non-institutionalized US population, while the BLSA is a study of healthy aging adults.

The UK Biobank study protocol differed in key ways from the studies using NHANES and BLSA data regarding device placement and wear-time protocols. The UK Biobank study used a tri-axial wrist-worn accelerometer, implemented a 24 h wear-time protocol, and summarized the raw acceleration data using the open source summary Euclidean Norm Minus One. In contrast, the NHANES 2003–2006 study had participants wear an uni-axial accelerometer at their waist, implemented a wake-wear protocol where participants were instructed to remove the accelerometer while sleeping, and summarized the raw acceleration data in one-minute epochs using a proprietary “activity count” summary. The BLSA study used a uni-axial chest-worn accelerometer, implemented a 24 h wear-time protocol, and summarized the raw acceleration data in one-minute epochs using a proprietary “activity count” summary. Note that the activity counts reported from the devices used in the NHANES 2003–2006 and BLSA studies are not directly comparable.

## 3. Results

Our analysis examines weekday (Monday–Friday) activity profiles and binary active versus inactive profiles across ages for 88,797 men and women in the UK Biobank accelerometer study. In the first part of our analysis we apply functional regression to the activity profiles, providing insight into how patterns in magnitude of acceleration vary across ages and gender. These results are shown in Figure 2 and Figure 3. The second part of this analysis applies both functional regression and curve registration to the binary activity profiles, providing information on age and gender-related differences in the probability of being active at each time of day as well as age and gender-related shifts in average timing of physical activity. These results are shown in Figure 4 and Figure 5.

Figure 2 shows the population estimated average acceleration by time of day over the age range in our analytic sample separately for males (panel A), females (panel B), and the difference between males and females (panel C). The color in the panels A/B of Figure 2, corresponding to average activity patterns in males and females, respectively, denotes lower (dark/light purple) versus higher (light/dark green) acceleration (volume of PA). Color in panel C of Figure 2 indicates periods of time where men are more (green) or less (purple) active than women and a solid black line demarcates areas where the estimated difference crosses 0 (i.e., no difference in activity levels between men and women, white color). We observe that among both males and females, younger participants tended to start activity earlier in the day (06:00 a.m. for 45–55 year olds vs. 07:00 a.m. for 65+ year olds) and maintain higher levels of activity later in the day, on average. In addition, with the exception of the youngest males in the sample, the peak average activity in this population occurs in the morning.

We also see a clear shift in activity patterns beginning around age 60 that manifest similarly in males and females; specifically, these individuals start their activity later in the morning, and tend to wind down earlier in the afternoon, consistent with previous findings in US populations, specifically NHANES 2003–2006 [38] and BLSA [22]. In addition, while both males and females under the age of 60 have two clear peaks in activity (around 8–9 a.m. and 6–7 p.m.), there is only one clear peak (around 10–11 a.m.) in older men and women (age ≥ 60). This change from two peaks to one peak after age 60 can also be seen in the BLSA study from Figure 3 of [22]. This clear shift in activity patterns around the age of 60, seen in both US studies and now the UK Biobank study, may be a result of individuals beginning to exit the work force via retirement, leading to a change in the structure of individuals’ weekday schedules.

In the period roughly between 10 a.m.–12 p.m. and 2 p.m.–4 p.m. in males, and, to a lesser extent females, average activity dips in the younger individuals (ages 45–60). The 2 p.m.–4 p.m. dip in PA has been previously observed in the BLSA [22], but the 10 a.m.–12 p.m. dip has not been previously reported. It may be that the larger sample size of the UK Biobank enables detection of more nuanced patterns than in smaller studies, or it may be specific to the UK population. Alternatively, the observed shift may be due to differences in the ENMO summary measure as compared to proprietary activity counts used by Xiao et al. [22] generated by similar activities, the different location of the device (wrist versus hip/chest), or some combination of the two. Because this pattern may be a result of structured lunch breaks for employed individuals, we term it the “lunch effect”.

Moving to the right panel of Figure 2, we see that across the age range of this sample, women tend to be more active than men during the daytime hours of 6 a.m.–6 p.m. (mostly light/dark purple color during this period) with the exception of 12 p.m.–1 p.m. (white/light green color), and less active during the nighttime hours of 12 a.m.–6 a.m. (mostly light green color). Interestingly, younger women (ages 45–60) tend to be less active during the evening hours of 6 p.m.–12 a.m., but older women (ages 60–80) are more active during this period. The observed higher level of average activity in males as compared to females in the early morning hours could be driven by poorer sleep quality (more movement during sleep), more variable sleep periods, or less stable weekday circadian patterns in men.

To give perspective on the average total amount of activity accumulated over the 12 a.m.–12 a.m. period, Figure 3 plots the estimated cumulative average activity at ages 50 (black line), 60 (red line), an 70 (blue line) separately for males (panel A), females (panel B), and the difference between males and female (panel C). Solid lines present point estimates and dashed lines represent 95% confidence intervals obtained via bootstrap. Consistent with the results from Figure 2, we see that younger individuals accumulate as much or more activity at any point of the day as compared to older individuals (black curve ≥ red curve ≥ blue curve at all times of the day). Comparing the estimated cumulative activity curves for age 50 versus 60 in both males and females, we see that the younger age group accumulate more activity early in the morning (larger difference between the two curves between 6 a.m. and 10 a.m.) which shrinks to near zero difference moving into the early afternoon due to the aforementioned “lunch effect” for younger adults, separating again in the late afternoon/early evening, resulting in more overall activity for the younger group. In addition, from Figure 3 panel C, we see that the estimated total daily activity for females and males is roughly equal for the younger ages groups as the confidence intervals at 12 a.m. just overlap 0 (95% confidence intervals for ages 50 and 60 contain 0), while older (age 70) females have noticeably higher estimated levels of total activity than older males (point estimate for age 70 at 12 a.m. is negative and the 95% confidence interval does not contain 0). This suggests that the lower levels of activity observed in men ages 50–60 during the day seen in the right panel of Figure 2 are offset by increased activity during the evening and early morning hours. In contrast, by age 70, the increased activity of males during the early a.m. hours is not enough to “make up” for the higher levels of activity in females during the rest of the day. These results find that in this population women are estimated to be as or more active than men ages 45–80 with women being relatively more active after age 60. This may indicate that activity patterns changed deferentially by gender as individuals begin to exit the workforce as they approach retirement age, though future investigations to validate this hypothesis are required.

Shifting focus to population level diurnal probabilities of being active versus inactive, consider Figure 4. The interpretation and layout of the figure is the same as Figure 2 except that color intensity now denotes probability of being active (or, in the right panel, difference in probability of being active). We find that the overall trends are largely the same as those seen in Figure 2 and Figure 3 with a few key differences. Regarding similarities, from the left two panels of Figure 4 we see that the estimated probability of being active tends to be highest in the morning shortly after waking, that this peak occurs later in older individuals, that there are morning and early evening peaks in younger males and females, and that there is a “lunch effect” that is present in both men and women. Analyzing the binary activity profiles provides additional information that was not revealed by the analysis of participants’ continuous activity profiles. Specifically, from the panel C of Figure 4, we see that females tend to have a higher probability of being active during the daytime hours (6 a.m.–6 p.m.) across all age groups and a lower probability of being active during the early a.m. hours (12 a.m.–6 a.m.). Interestingly, although males ages 45–60 were generally found to have higher levels of average activity 12 p.m.–2 p.m. and 6 p.m.–12 a.m., women have nearly the same or higher estimated probability of being active. This suggests that women are, on average, engaging in more low-light levels of activity and less moderate-vigorous intensity activity during this time period. Alternatively, the average volume of PA may be pulled upward by a few individuals engaging in very vigorous levels of activity, though future work is needed to confirm this theory.

Finally, consider Figure 5, which plots the estimated average warping functions for males (panel A), females (panel B), and the difference between males and females (panel C) for ages 50 (black line), 60 (red line), and 70 (blue line) years old. From panels A/B of Figure 5 we see that in the morning period, the ordering of the lines is black (age 50) greater than red (age 60) greater than blue (age 70). This pattern reverses after around 10 a.m. in both men and women. This ordering of the curves is consistent with a compressed chronological “active” time during the daytime hours and an expanded “inactive” time during the nighttime/early morning hours, with active time being more compressed with increasing age. We also see this phenomena in Figure 2 and Figure 4. However, the observed differences in warping functions are visually quite small. Looking at panel C, we see that for the youngest age group shown (red line, age 50), the estimated difference is very close to zero at all times of day, indicating similar active/inactive timing between men and women age 50. In contrast, for the next youngest age group (red line, age 60), the estimated warping function is estimated to be positive for the entirety of the day, suggesting that men’s active periods are uniformly shifted to earlier in the day relative to women age 60. Finally, for the older age group (blue line, age 70), the estimated warping function starts negative and then becomes positive after around 8 a.m., suggesting men have a compressed active period relative to women age 70. These observations are all consistent with the patterns observed in Figure 2 and Figure 4. However, once again, the estimated differences in average warping functions do not appear large in absolute value.

## 4. Discussion

This work extends the existing literature on physical activity and aging through the (1) validation of aging trends seen in several US populations in a large UK cohort; (2) establishment of functional regression approaches as a set of computationally feasible, open source tools for analyzing physical activity in the largest publicly available accelerometry dataset; (3) combination of functional regression and registration for analyzing the associations among age, gender, and the timing and the volume of physical activity; (4) analysis of multiple types of PA acceleration profiles (activity count versus binary activity profiles) which showed that conclusions based about PA and aging are potentially dependent on profile choice; and dissemination of template code which allows researchers to reproduce our analytic procedure on any accelerometry dataset via an application to the NHANES 2003–2006 accelerometry data. In the future, the tools presented here could be used to create reference quantities for normative patterns of physical activity by jointly considering multiple types of PA profiles. Further, the methods applied in this study could be applied to other bio-signals measured continuously using wearable devices which have circadian patterns; for example, heart rate, continuous glucose monitoring, or skin temperature.

This study has several important limitations. First, we reduced multiple days of subjects’ accelerometry data to one day per subject by collapsing across days. It may be possible to gain additional insights into the epidemiology of the timing and volume of physical activity and aging by analyzing individuals’ day-level data using a multi-day approach to both registration and regression. We also restricted our analysis to weekdays only. It may be that analyzing individuals weekend activity patterns, when many employed people’s activity are not restricted by their occupational requirements, could provide additional insights into individuals’ leisure time activities. Moreover, the registration method we applied here was not designed to handle individuals who tend to sleep during the daytime. That is, registration does not respect the circular nature of clock time.

In addition, the composition of the UK Biobank accelerometry sub-study has been shown to be overall healthier than the larger UK Biobank study, which is itself healthier than the UK population. As a result, the generalizability of our results is unclear. Nevertheless, we were able to replicate key findings regarding the epidemiology of circadian patterns of PA from previous studies in both nationally representative and healthy aging US cohorts, suggesting that the observed aging trends are robust to some sources of sampling bias and are likely to be reproduced in other studies which draw from similar populations. Finally, driving and other systematic behavioral differences that are not physical activity but affect accelerometry readings across age ranges and sexes can have a minor effect on results, so all results must be viewed with this caveat [39]. Threshold-based methods such as our binary-curve registration are more robust.

Synthesizing the results of our analysis, we find that the diurnal patterns of both volume of physical activity and the probability of being active change with age and that there are sizable gender difference in these trends. In addition, there does appear to be an age trend in the timing of PA as older adults have a more compressed “active” period. However, there does not appear to be a substantial gender difference in the changes of the active period with age. Ultimately, from a scientific perspective, this study validates previous studies’ findings in a new aging cohort (diurnal patterns of volume), presents novel findings regarding the difference in analyzing various summaries of physical activity profiles (probability of being active/inactive, registration and changes in phase in timing of PA). In addition, we introduce new methodologies to the study of PA and aging and, crucially, provide the code for reproducing our methods using publicly available software through the accompanying online supplemental material to be uploaded to Github on publication.

## Figures and Tables

**Figure 1 sensors-21-01545-f001:**
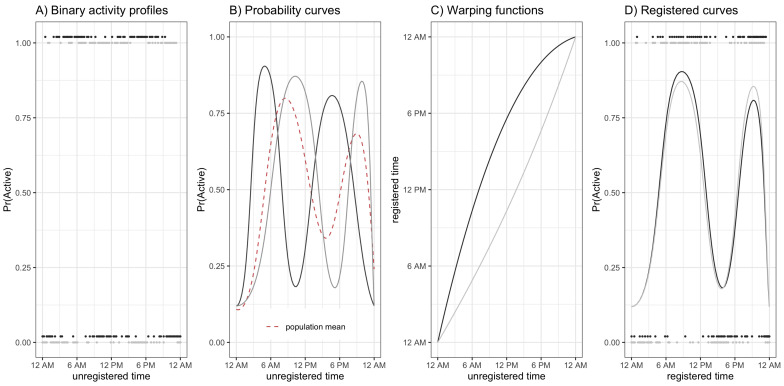
Toy Registration Example. Black and gray colors denote two separate subjects, and panels represent (**A**) binary activity profiles, (**B**) unregistered probability curves, (**C**) warping functions, and (**D**) binary activity profiles and probability curves that have been aligned using the warping functions in panel (**C**).

**Figure 2 sensors-21-01545-f002:**
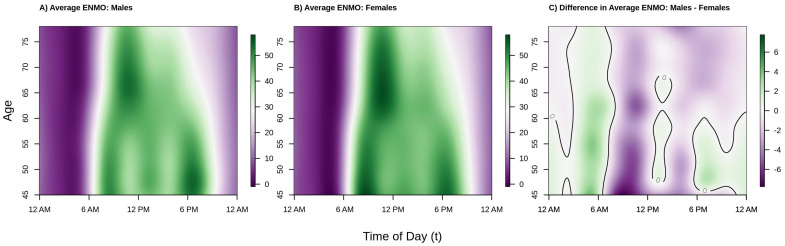
Estimated population average acceleration (ENMO) by age and time of day for Males (**panel A**), Females (**panel B**), and the estimated difference between Males and Females (**panel C**). Color intensity in panels A/B indicates lower (purple) versus higher (green) levels of activity. Color intensity in panel C indicates Males are less (purple), the same (white), or more (green) active as compared to Females, with a solid black line demarcating the transitions between regions where males are more or less active than females (i.e., the difference is 0). Estimates of these surfaces in panels A/B are obtained by fitting a function-on-scalar regression (FoSR) model separately by sex where the population average ENMO is allowed to vary smoothly in both time of day and age. The FoSR model is fit by modelling the average ENMO as a tensor product smooth of marginal spline bases, with cyclic cubic regression splines used in the time domain to respect the cyclic nature of time. Panel C is obtained by taking the difference of panels A and B.

**Figure 3 sensors-21-01545-f003:**
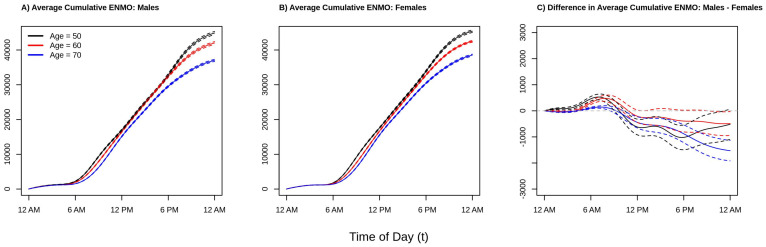
Estimated population cumulative average acceleration (ENMO) by time of day for Males (**panel A**), Females (**panel B**), and the estimated difference between Males and Females (**panel C**). separately for ages 50 (black), 60 (red), and 70 (blue) years old. Solid lines denote point estimates and dashed lines represent point-wise confidence intervals. Estimates of the curves presented in panels A/B are obtained by fitting a function-on-scalar regression (FoSR) model separately by sex where the population average activity count is allowed to vary smoothly in both time of day and age, then numerically integrating the estimated population average activity count over the 12 a.m.–12 a.m. period. The FoSR model is fit by modelling the average ENMO as a tensor product smooth of marginal spline bases, with cyclic cubic regression splines used in the time domain to respect the cyclic nature of time. Panel C is obtained by taking the difference of panels A and B.

**Figure 4 sensors-21-01545-f004:**
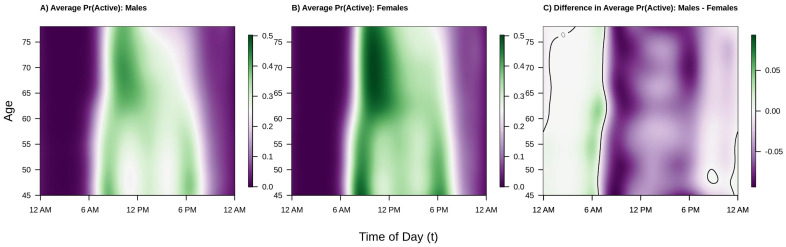
Estimated population average probability of being active (or, more specifically, generating ENMO greater than or equal to 30 milli-gravity units) by age and time of day for Males (**panel A**), Females (**panel B**), and the estimated difference between Males and Females (**panel C**). Color intensity in panels A/B indicates lower (purple) versus higher (green) probability of being active. Color intensity in panel C indicates Males are less (purple), the same (white), or more (green) likely to be active as compared to Females, with a solid black line demarcating the transitions between regions where males are more or less likely to be active than females (i.e., the difference is 0). Estimates of these surfaces in panels A/B are obtained by fitting a generalized function-on-scalar regression (FoSR) model separately by sex where the log odds of being active is allowed to vary smoothly in both time of day and age. The FoSR model is fit by modelling the log odds of being active as a tensor product smooth of marginal spline bases, with cyclic cubic regression splines used in the time domain to respect the cyclic nature of time using binary active/inactive profiles. Panel C is obtained by taking the difference of panels A and B.

**Figure 5 sensors-21-01545-f005:**
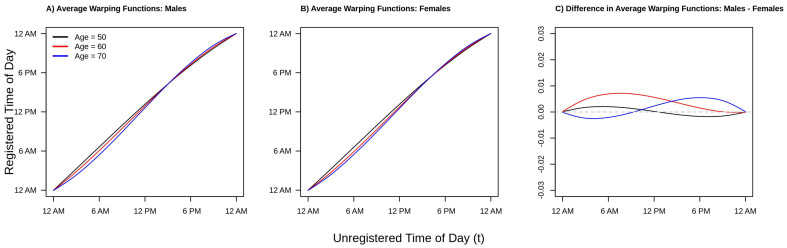
Estimated population average warping functions for Males (**panel A**), Females (**panel B**), and the estimated difference between Males and Females (**panel C**). separately for ages 50 (black), 60 (red), and 70 (blue) years old. Estimates of the lines presented in panels A/B are obtained by fitting a function-on-scalar regression (FoSR) model separately by sex where the population average warping function is allowed to vary smoothly in both time of day and age. The FoSR model is fit by modelling the average warping function as a tensor product smooth of marginal spline bases with cubic regression splines used in both the time and age domains. Panel C is obtained by taking the difference of panels A and B.

## Data Availability

The accelerometer datasets generated by UK Biobank analyzed during the current study are available via the UK Biobank data access process. (see http://www.ukbiobank.ac.uk/register-apply/ accessed on 1 January 2021).

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
