# Peer review of "Diurnal Physical Activity Patterns across Ages in a Large UK Based Cohort: The UK Biobank Study"

_sensors, 2021, doi:10.3390/s21041545_

Round 1

Reviewer 1 Report

The manuscript describes the application of curve registration for diurnal activity profiles recorded by body-worn accelerometers. The essential strength of the presented research is that it investigates physical activity captured by accelerometers located on the wrist (increasingly used sensor placement) over an enormous population of over 80000 individuals monitored in free-living conditions. Another benefit of the described methodology is that it appears to be computationally efficient, while the released source code allows to reproduce findings in different populations. However, before I make any recommendations, I feel the manuscript needs few improvements listed below.

(Introduction) Please rephrase sentence beginning in line 62.

(Dataset) Explain why you focused on population specifically above 42 years of age.

(Dataset) When was the data collected (for reference to e.g., NHANES)?

(Dataset) Briefly, what is the protocol to aggregate 5-second epochs into 1-minute epochs (line 111)?

(Dataset) Briefly, what is the protocol behind wear time detection?

(Dataset) Briefly, what are the three data quality criteria?

(Dataset) “88933 participants” out of? What was the total cohort size and how did the inclusion/exclusion criteria affect on the selected subsample?

(Notation) It might be somewhat confusing to swing between “proprietary activity counts” (lines 91, 228), “activity count profiles”(line 203,271), and “activity counts” (line 130, caption to Figure 3). Consider rephrasing.

(Notation) “g” represents both units of acceleration and a link function. Consider changing.

(Figure 2) The change is daily activity profiles in males (<60: more active afternoons vs. >60: more active mornings) should be highlighted more.

(Figure 4) Sentence: “that there is a “lunch effect” that is more pronounced in men than in women”. It does not seem so from the figure.

(Figure 5) Panels A and B are illegible. Also, it is not clear how to interpret panel C, e.g., what’s the interpretation of warping function equal to 0.0, 0.5, 1.0?

(Results/discussion) Sentence beginning in line 244: specify what you mean by “younger individuals” and “older individuals”

(Results/discussion) Sentence beginning in line 314 is confusing, consider rephrasing.

(Results/discussion) Provide examples of other bio-signals that you have in mind in line 303.

Reviewer 2 Report

Summary: The paper uses the UK biobank study to analyze the diurnal activity patterns of the population in the UK. The study is well motivated and the analysis performed in the paper is useful from a public health perspective.

Comments:

  • The authors use acceleration as an indication of physical activity. However, users may experience acceleration due to other effects as well. For example, driving or being in a public transportation vehicle can register acceleration. It would be great if the authors can comment on this and also discuss how these artifacts can be handled in the analysis.
  • The discussion section states that some new methods of data analysis are presented in the paper. However, it is not clear what are the new methods. The reviewer suggests the authors to clearly identify the new methods for the benefit of the readers.
  • Please comment on how the results will change if multi-level activity Y_i (t) function is used. That is, instead of a binary classification, we can include four levels of activity intensity (e.g. walking v/s running). Will such an analysis offer more insights into the activity patterns?
  • As I understand, one of aims of the paper was to compare the aging trends in PA in different populations across countries. However, no such comparison is presented in the paper. Please include some analysis to this effect so that results from UK population can be put in perspective in terms of other countries and also globally.

Round 2

Reviewer 1 Report

Thank you for addressing my comments. I think the incorporated changes have improved the clarity of data processing and analysis descriptions as well as interpretation of results. Once the final results from the entire dataset are provided (Figure 5), I will to recommend this manuscript for publication.

Some minor text corrections comments, like in line 327 (“with with”), line 324 (“we that”), line 107 (“years” instead of "days").

Reviewer 2 Report

The authors have addressed all my questions in the review response. As a follow up, it would be great if the authors update all the results with the updated samples in the final version of the paper. Even though the results are similar, it would benefit readers to have results from the same set of samples.
